# On Mitigating the Effects of Multipath on GNSS Using Environmental Context Detection

**Arif Hussain** [1,*,†], **Arslan Ahmed** [2,3,†], **Madad Ali Shah** [2], **Sunny Katyara** [2], **Lukasz Staszewski** [4] and **Hina Magsi** [2]

1   Department of Electronics Engineering Technology, The Benazir Bhutto Shaheed University of Technology & Skill Development, Khairpur Mirs 66020, Pakistan
2   Department of Electrical Engineering, Sukkur IBA University, Sukkur 65200, Pakistan
3   Nuclear Advanced Manufacturing Reserach Center (NAMRC), Rotherham S60 5WG, UK
4   Department of Electrical Power Engineering, Wroclaw University of Science & Technology, 50370 Wroclaw, Poland
*   Correspondence: arif@bbsutsd.edu.pk
†   These authors contributed equally to this work.

**Abstract:** Accurate, ubiquitous and reliable navigation can make transportation systems (road, rail, air and marine) more efficient, safer and more sustainable by enabling path planning, route optimization and fuel economy optimization. However, accurate navigation in urban contexts has always been a challenging task due to significant chances of signal blockage and multipath and non-line-of-sight (NLOS) signal reception. This paper presents a detailed study on environmental context detection using GNSS signals and its utilization in mitigating multipath effects by devising a context-aware navigation (CAN) algorithm that detects and characterizes the working environment of a GNSS receiver and applies the desired mitigation strategy accordingly. The CAN algorithm utilizes GNSS measurement variables to categorize the environment into standard, degraded and highly degraded classes and then updates the receiver's tracking-loop parameters based on the inferred environment. This allows the receiver to adaptively mitigate the effects of multipath/NLOS, which inherently depend upon the type of environment. To validate the functionality and potential of the proposed CAN algorithm, a detailed study on the performance of a multi-GNSS receiver in the quad-constellation mode, i.e., GPS, BeiDou, Galileo and GLONASS, is conducted in this research by traversing an instrumented vehicle around an urban city and acquiring respective GNSS signals in different environments. The performance of a CAN-enabled GNSS receiver is compared with a standard receiver using fundamental quality indicators of GNSS. The experimental results show that the proposed CAN algorithm is a good contributor for improving GNSS performance by anticipating the potential degradation and initiating an adaptive mitigation strategy. The CAN-enabled GNSS receiver achieved a lane-level accuracy of less than 2 m for 53% of the total experimental time-slot in a highly degraded environment, which was previously only 32% when not using the proposed CAN.

**Keywords:** GNSS; multipath traversing; context detection; adaptive mitigation

## 1. Introduction

The availability and accuracy of Positioning, Navigation and Timing (PNT) services can play a significant role in making transportation (road, rail, air and marine) more efficient, safer and sustainable by enabling route optimization and fuel economy planning [1–4]. Real-world applications, such as precision agriculture, real-time tracking, asset management, emergency response services, disaster management and autonomous driving, rely heavily on accurate positioning and time estimations [5–9] for proper functioning.

In addition, positioning and navigation technologies have recently played pivotal roles in the war against the COVID-19 pandemic by enabling the real-time tracking, tracing and locating of virus hot-spots by deploying geo-fencing [10–12] to monitor and control the

spread of COVID-19. Among the various available solutions for precise positioning and navigation [13–16], the Global Navigation Satellite System (GNSS) has established itself as a globally dominant and cost effective navigation technology [14,17].

Positioning and navigation in urban canyons has always been a challenging task [18–20] as GNSS receivers encounter areas with an insufficient availability of satellites due to multipath (MP) and/or non-line-of-sight (NLOS) signal reception. This remains a potential vulnerability of satellite-based navigation systems due to biased range measurements and inaccurate positioning estimates [21,22].

In [23], an initial study was performed on the characteristics of GNSS signals in highly obstructed environments by detecting and analyzing the possible deviations due to MP and NLOS reception. The quality of GNSS signals was analyzed and compared under different environments (i.e., LOS and NLOS) through field experiments. The results of these experiments showed that MP/NLOS could significantly affect the GNSS signal in terms of degraded tracking performance and frequent loss of lock.

At present, there are four independent global satellite constellations (GPS, GLONASS, Galileo and BeiDou) with more than 100 units in orbit that are transmitting signals at several frequency bands to increase the satellite density [24]. This helps to improve the positioning availability and accuracy in constrained and highly degraded multipath environments. This was also demonstrated in [25] where it was shown that, when combining four constellations (GPS, GLONASS, Galileo and BeiDou), more than 35 satellites could be accessed and tracked at a given location.

Similarly, researchers at Taoglas Inc. USA demonstrated a noticeable improvement in the positioning accuracy when using dual constellations (GPS and GLONASS) compared to single constellations [26]. The European GNSS Agency (ESA) and *Rx* Networks Inc. performed several performance assessment tests in real-world environments and demonstrated that Galileo, when combined with GPS, provided more accurate positioning estimates compared to the GPS-only configuration [27].

In [28], researchers demonstrated that the performance of a tri-constellation navigation system in an obstructed environment (e.g., medium-height buildings) matched the performance of a GPS-only system in a clear open-sky environment when using only those satellites with elevation angles greater than 32 degrees. This implies that the impact of obstructions below 32 degrees can be mitigated by using an increased number of constellations.

Jing Guo et al. [5] demonstrated that, in degraded environments (e.g., farms surrounded with dense trees), the availability and accuracy of positioning decreased significantly for the GPS-only system; however, limited impacts were observed for the multi-constellation GNSS case due to a greater number of visible satellites. This was experimentally validated in [25], where the impacts of using multiple constellations compared to a single constellation system in three distinct environments (i.e., open sky, partially obstructed and highly obstructed environments) were investigated.

The results of our previous work on multi-constellation GNSS suggested that the positioning performance of a GNSS receiver may vary significantly with the type of operating environment. It has been established that the use of multiple constellations can enhance the availability and reliability of GNSS in constrained environments; however, this cannot guarantee improved positioning accuracy as debated in [5,25–28].

The positioning accuracy of satellites is usually determined by the availability, geometry and the quality of the received signals, and these factors are highly influenced by the dynamics and the structure of the urban canyon. At first, the obstructions in urban areas may result in blockages of the line-of-sight (LOS) of satellites, and this eventually distorts their geometric distribution [21]. However, this can be compensated for by using a multi-constellation GNSS configuration.

Secondly, the flat surface reflectors give rise to multipath (MP) or non-line-of-sight (NLOS) phenomena, which further degrade the quality of signals by introducing biases in the range measurements. There exists substantial research for improving the GNSS

performance by minimizing the impact of MP and/or NLOS reception. This includes techniques to detect, model and mitigate the NLOS/multipath effects at various aspects, including the antenna, receiver and measurement [21,29,30].

Furthermore, there is another class of multipath/NLOS mitigation techniques that works at the discriminator level, such as the maximum likelihood techniques based on the multipath estimating delay locked loop (MEDLL) [31–33], the coupled amplitude DLL (CADLL) [30] and the multipath insensitive DLL (MIDLL) [34]. However, these mitigation approaches require an increased number of correlators, which results in increased system complexity and computational load.

Recently, the mitigation techniques at the receiver and measurement levels (i.e., detecting, de-weighting and rejecting the affected range measurements) has received significant attention because they do not require major hardware modifications [29]. These mitigation models/techniques, however, work effectively only in some contexts and not in all operating environments.

Similarly, the correlator-based techniques (i.e., MEDLL or its variants) cannot detect short-delay multipath environments and may introduce bias into the measurements due to their modifications inside the tracking channels of a correlator [30,35]. Typically, the detection and exclusion of faulty measurements (i.e., signals affected by MP/NLOS) help to improve the positioning accuracy; however, this leads to outages in dense urban environments due to the deficiency of redundant range measurements [25].

Practically, GNSS receivers operate in the wide range of environmental contexts, which significantly influences the signal reception conditions and severity of multipath/NLOS reception. Similarly, most of the multipath mitigation methods are not capable of effectively suppressing the NLOS/multipath effects in all contexts. Hence, in order to operate accurately and effectively in a wide range of environments, a GNSS receiver is required to adopt an optimal mitigation technique/method based on the detected context. Utilizing an inappropriate method may result in less effective reduction of multipath effects while consuming more power with a greater computational load.

There exist several methods for environmental context detection and characterization using the GNSS parameters, i.e., satellite availability, DOP, residuals and signal strength or its variants [36–42],to detect and identify the type of environmental context.

Most of the previous work focused on the development of context-detection models using single and dual constellations with little to no consideration to how these models can be utilized at the at receiver level to improve the availability and accuracy of navigation services [36–42].

In almost all of the previous work on context detection, signal strength or its variants were used as fundamental environment recognition parameters; however, this may not result in accurate context detection in the case of multi-constellation GNSS because: (1) the strength of the received signal is highly affected by NLOS and/or multipath; however, the severity of effects varies by frequency [43]; (2) the signal strength is majorly influenced by the elevation angle and can also be affected by the receiver efficiency or antenna design; and (3) a combination of multiple navigation systems in multi-constellation GNSS mode results in increased satellite density, and therefore monitoring the strength of each satellite can lead to a huge processing load.

This paper presents a detailed study on environmental context detection using multi-frequency and multi-constellation GNSS signals. A new context-aware navigation (CAN) method is proposed with context detection and characterization capabilities to mitigate the multipath effects in constrained environments by using adaptive GNSS receiver design in the multi-constellation mode.

The ultimate aim of work is to improve the navigation accuracy by detecting the operating environments and then adjusting the mitigation strategy accordingly. The CAN method utilizes a handful of GNSS parameters to categorize the environment into standard, degraded and highly degraded classes and then updates the receiver tracking loop based on the inferred environment to suppress the degradation effects.

The main difference between the proposed CAN and other methods is that it does not use the signal strength or its variants as key parameters for environment detection, instead it utilizes the globally and equally distributed property of GNSS, i.e., satellite availability and a new feature named the change factor (CF) for environment categorization. Additionally, the efficient design of proposed CAN, which includes selection of the optimal GNSS parameters along with context definitions for environment characterization makes it more practically realizable as compared to complex methods that would become practically impossible to implement in receivers.

Finally, to validate the performance of the proposed CAN algorithm, a detailed study on a multi-constellation GNSS receiver in the quad-constellation mode, i.e., GPS, BeiDou, Galileo and GLONASS, is conducted by driving an instrumented vehicle in and around the city center, and the results are then compared with a standard receiver. The rest of the manuscript is organized as follows: Section 2 discusses the methodology used for GNSS performance evaluation in a multi-constellation mode with field experiments, route dynamics, observation periods and signal-reception characteristics. Section 3 highlights the results of GNSS performance evaluation along with the statistical characteristics of the received signal quality. Section 4 presents the context-aware navigation (CAN) algorithm and its implementation on a real-time GNSS receiver. Finally, Section 5 gives our concluding remarks and recommendations for future extensions.

## 2. Performance Evaluation of a Standard Multi-Constellation GNSS Receiver

As we increasingly rely on GNSS-based positioning, understanding and mitigating GNSS vulnerabilities has become a critical risk management activity for manufacturers, systems and application providers as well as end-users as this can impact national security and may bring huge economic losses. The rigorous performance assessment under realistic conditions is key to understanding the different types of GNSS vulnerabilities and error contributors. In many studies, the performance of a GNSS receiver has been evaluated and characterized through field experimentation in static mode for long durations [44–49] or under controlled environmental conditions/settings [50].

Static tests provide good analysis of GNSS errors and can be used to establish cause and effect relationships; however, they are generally performed on pre-surveyed and known candidate sites. In addition, these tests provide little information regarding inaccuracies in positioning that occur due to movement or dynamics of the environment. On the other hand, on-road/field tests can be performed while navigating through different environmental contexts for more realistic GNSS performance assessment in a dynamic mode.

However, these tests are demanding in nature and require a considerable skill set and amount of subject knowledge to select accurate experimental sites, route trajectories and operational modes/scenarios [24,51–54].

In this section, dynamic performance assessment of a multi-constellation and multi-frequency standard-mode GNSS receiver is performed using fundamental quality indicators, such as the satellite availability, Position Dilution of Precision (PDOP) and Root Mean Square (RMS) error in congested city center areas. For this assessment, all four global constellations, i.e., GPS, BeiDou (BDS-3), Galileo and GLONASS, are utilized for the first time without using any augmentation or mitigation services.

Although there exist several studies evaluating the on-road performance of GNSS and a detailed comparison of these on-road analyses can be found in Section III of [24]. However, a majority of previous studies have been conducted on dual or triple constellations with GNSS correction services (i.e., Networked RTK, Proprietary) enabled. In order to highlight the pros and cons of using an increased number of constellations for positioning estimation, the next section covers the details of the field experiments performed, route dynamics, observation periods and signal-reception characteristics.

### 2.1. Experimental Sites/Route for Performance Evaluation

The route used for dynamic performance evaluation is a 30 km long route that covers the center and surroundings of Sukkur city in Pakistan i.e, including residential and industrial areas as well as highways. The route contains a wide range of operating environments, such as clear open-sky at the N-65 highway, semi-urban and dense urban areas with small lane widths, high-rise buildings, flyovers and bridges and thus is a more realistic test-run for urban canyons. For detailed performance assessment, the experimental route is divided into four distinct observation windows labeled as P1, P2, P3 and P4 as shown in Figure 1.

The observation windows are selected based on the type of environment and obstacles on the vehicle route. The first observation window, P1, in Figure 1 is 4 km long, which is an N-65 highway connecting Sukkur to other cities. The P1 is a two-lane one-way route with mostly clear open-sky views resulting in excellent radio signal reception. The second observation window, P2, is a 1.6 km long bridge on the river Indus known as Lloyd's Barrage. At P2, there are good chances of signal degradation (reflection or blockage) because of the structure of the bridge as shown in Figure 1.

Then, a 4 km long route at the center of Sukkur city is taken as the third observation window (P3). P3 is a densely populated area with high-rise buildings, congested roads, flyovers and two cantilever bridges (Lansdowne Bridge) each 310 feet long, where there are significant chances of blockage and signal quality degradation due to high multipath/NLOS reception. Finally, a 5 km long route with little or no obstruction is taken as the last observation site and labeled as P4.

This site has similar characteristics compared with those of P1. The route length, observation time and signal-reception characteristics of each of the observation windows are given in Table 1. Furthermore, the details of the field experimentation, which include the equipment used, number of constellation utilized, total time duration of each study and the antenna height, are given in Table 2.

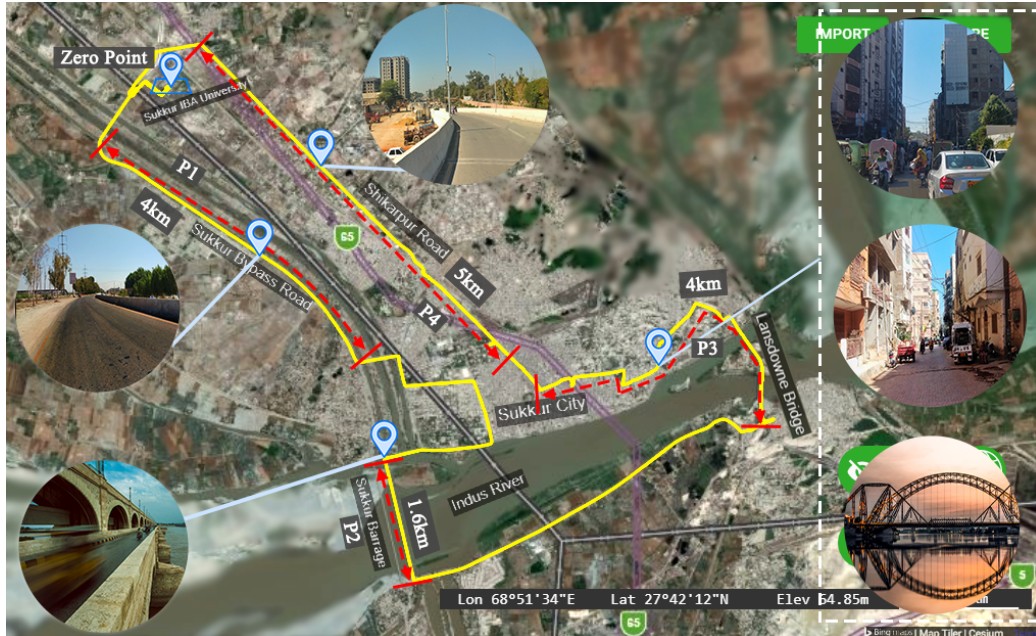

**Figure 1.** A 30 km long route covering the Sukkur city center and its surroundings used for field experimentation contain various operating environments, such as clear open-sky at the N-65 highway, semi-urban and dense urban areas with small lane widths, high-rise buildings, flyovers and bridges. The total route is divided into four distinct observation windows based on environment characteristics, and these are labeled as P1, P2, P3 and P4.

**Table 1.** The route length, observation time and signal-reception characteristics of the observation windows P1, P2, P3 and P4.

| Site | Location | Route Length [km] | Time [min] | Characteristics |
|------|----------|-------------------|------------|-----------------|
| P1 | N-65 Highway | 4.0 | 02–10 | Mostly open-sky views |
| P2 | Lloyd Barrage Sukkur | 1.6 | 14–22 | Partial obstruction |
| P3 | Center of City | 4.0 | 37–62 | Severe obstruction |
| P4 | Shikarpur Road | 5.0 | 65–80 | Mostly open-sky views |

**Table 2.** Equipment, duration, constellations and the antenna height for the two experiments.

| Experiment | Antenna | Receiver | Constellations | Experiment Duration | Antenna Height |
|------------|---------|----------|----------------|---------------------|----------------|
| Exp1 (without CAN) | PolaNt Choke Ring B3/E6 | Septentrio PolaRx5S | G+R+E+C [1] | 11-09-2021 80 (min) | 1675 mm |
| Exp2 (with CAN) | PolaNt Choke Ring B3/E6 | Septentrio PolaRx5S | G+R+E+C | 18-09-2021 80 (min) | 1675 mm |

[1] G = GPS, R = GLONASS, E = Galileo, C = BeiDou.

## 2.2. Experimental Setup and Data Collection

The experimental setup and test vehicle used for the performance evaluation are shown in Figure 2. The high-precision antenna (PolaNt Choke Ring B3/E6) was mounted on the roof of the vehicle and connected to a PolaRx5S multi-frequency, multi-constellation GNSS receiver. The PolaRx5S can acquire and track signals from all four constellations, i.e., GPS, BeiDou, Galileo and GLONASS. The field experiment was performed by driving the vehicle along the route shown in Figure 1. The vehicle started and ended its journey at the Sukkur IBA University with a total observation period of 80 min. The GNSS data was logged at a rate of 1 Hz. The technical specifications of the equipment, i.e., the receiver, antenna and supported frequency bands, are given in Table 3.

**Table 3.** Technical specifications of the equipment used.

| Equipment | Description/Technical Specification |
|-----------|-------------------------------------|
| PolaNt Choke-Ring B3/E6 Antenna | Polanet Choke ring is a high-precision geodetic multi-frequency multi-constellation antenna with the capability of receiving navigation signals from the entire GNSS Spectrum |
| Septentrio PolaRx5S | Multi-constellation and Multi-frequency GNSS Receiver |
| Hardware Channels | 544 |
| Sampling Rate | Up to 100 Hz |
| Supported Frequency bands | GPS (L1CA-1575.42 MHz, L2P-1227.60 MHz, L5-1176.45 MHz), GLONASS (L1CA-1602 MHz, L2C-1246 MHz, L3OC-1202 MHz), GALILEO (E1-1575.4 MHz,E5-1176.46 MHz,AltBoc-1191.79 MHz), BEIDOU (B1I-1561.9 MHz, B2I-1207.14 MHz, B3-1267.52 MHz). |
| Supported Data Formats | SBF (Septentrio Binary File), RINEX (Receiver Independent Exchange), ISMR (Ionospheric Scintillation Monitoring Record), BINEX (Binary Exchange Format), NMEA (National Marine Electronics Association) |

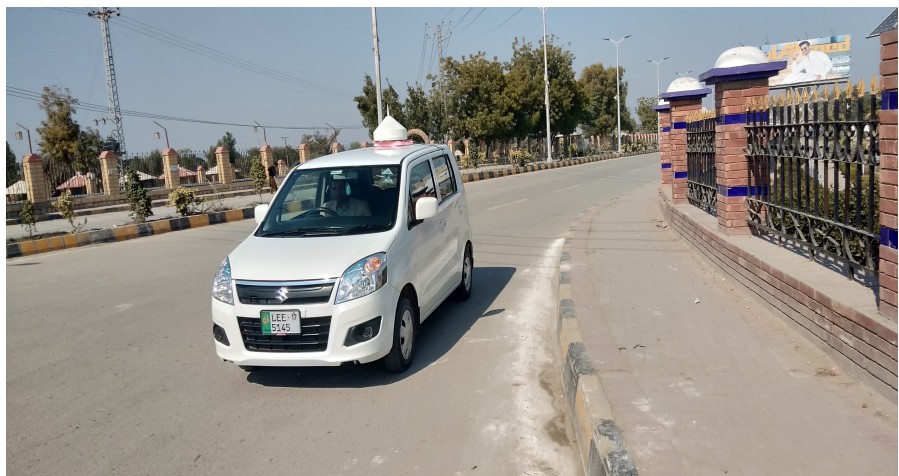

**Figure 2.** Instrumented test vehicle containing a roof-mounted antenna along with a PolaRx5s multi-frequency and multi-constellation GNSS receiver as used for field experimentation and data acquisition.

## 3. Performance Evaluation

The performance evaluation results of a GNSS receiver in quad-constellation mode in terms of fundamental quality indicators, i.e., the satellite availability, PDOP and RMS error over the entire experimental route of Figure 1 with a complete observation period of 80 min are shown in Figure 3. Figure 3a shows the number of tracked satellites, Figure 3b represents the PDOP, whereas the RMS error is shown in Figure 3c. As already mentioned, the experimental route comprises a real-world driving profile consisting of a clear open-sky, semi-urban and dense urban areas with small lane widths, high-rise buildings, flyovers and bridges.

The results in Figure 3 show that significant deviations may occur in the fundamental quality indicators (PDOP, RMS error, satellites in-view and average satellite availability) when navigating across different types of operating contexts. In observation window P1, an average of 34 satellites were locked by the receiver during the 8 min time-slot, and this was expected due to negligible obstruction at the route.

The availability of a large number of satellites ensures more accurate and seamless navigation because redundant measurements can help in selecting satellites with strong signal strengths and thus constitute good satellite geometry. In fact, excellent PDOP values (average PDOP < 1) were observed during this observation window as shown in Figure 3b. The RMS error in P1 is depicted in Figure 3c.

As expected, with adequate satellites in view and good geometry, the average RMS error was found to be 1.3 m with lower and upper bounds of 1 and 2.4 m, respectively. The results suggest that lane-level accuracy ($\leq$1.5 m) can be achieved at highways or suburban areas with standalone multi-constellation GNSS without any correction services.

As the test vehicle moved closer to the city center and passed through Lloyd's Barrage (observation window P2), the quality parameters began to deteriorate rapidly due to the surroundings (i.e., the structure of the bridge) resulting in multipath/NLOS environments. The average number of tracked satellites reduced to 27 from 34 (Figure 3a), and the maximum PDOP value increased to 2.21 in Figure 3b. Although the satellite availability was adequate at this point (i.e., a minimum of 17 satellites were acquired), the positioning performance was degraded.

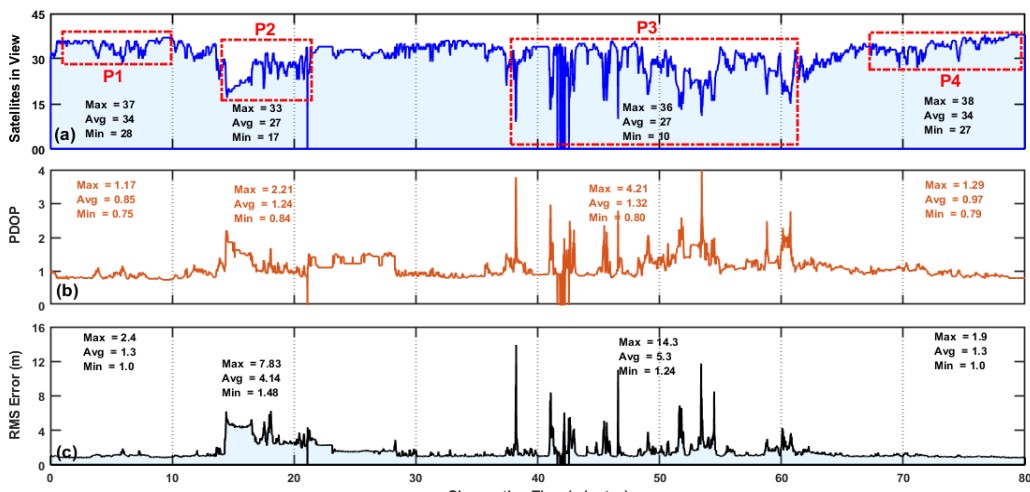

**Figure 3.** The performance evaluation of a standard GNSS receiver in quad-constellation mode in terms of fundamental quality indicators, i.e., (**a**) the satellite availability, (**b**) PDOP and (**c**) RMS error over the entire experimental route containing different operating environments denoted by P1, P2, P3 and P4.

The average and maximum RMS errors in Figure 3c for the P2 observation window were found to be 4.14 and 7.83 m, respectively; however, this level of accuracy is not sufficient in many cases. The degraded positioning performance in P2 was due to the multipath and NLOS signal reception because there was no significant drop in the PDOP value, which was less than 2 most of the time as observed in Figure 3b. The PDOP values indicate the impact of the spatial distribution of the satellites on the final accuracy of the positioning solution, and a scale of 0 to 10 is typically used to compare the severity of this effect.

A PDOP value of less than 2 indicates excellent satellite geometry, a value between 2 and 4 is acceptable, and PDOP values of greater than 4 refer to a poor geometrical distribution of satellites. Based on the PDOP values, it can be concluded that multi-constellation GNSS enhances the probability of better satellite geometry and lower PDOP values; however, the effects of multipath and NLOS signal reception lead to increased RMS errors in the positioning solution as can be seen in Figure 3c in the P2 observation window. This shows that the PDOP alone is not a good indicator for evaluating the GNSS performance.

In Figure 3, the third observation window, i.e, P3, highlights the GNSS performance in the dense urban areas with congested roads, high-rise buildings, towers and a cantilever bridge. Among all the observation windows, P3 showed greater deterioration in the number of tracked satellites, PDOP values and the RMS error.

Since the vehicle was continuously navigating and the GNSS antenna was experiencing surroundings with different obstruction levels, on average, 27 satellites were being tracked by the receiver, and this varied between 10 and 36 based on the obstruction level and multipath/NLOS signal reception during the entire P3 window as can be seen in Figure 3a. The average PDOP value in P3 was found to be 1.24 with rapid variations between 0.80 and 4.21. In Figure 3c, the maximum and average RMS errors were 14.3 and 5.3 m, respectively, which are large and may be problematic for the receiver to correctly navigate in many applications.

The last case, i.e., the P4 observation window, exhibited similar characteristics to P1 in terms of locked satellites, DOP and RMS errors due to similar environmental dynamics. It should be noted that, in this study, GNSS was used in standalone mode without any correction services or augmentation system. The results of the field experiments for the observation windows P1, P2, P3 and P4 for different types of environments are given

in Table 4, which gives a summarized overview of how the GNSS fundamental quality indicators can be affected while navigating through different environments encountering blockages, multipath environments and NLOS signal reception.

**Table 4.** Comparison of quad-constellation GNSS receiver performance under different types of environments denoted by the P1, P2, P3 and P4 observation windows.

| Site | Satellites in View max/avg/min | PDOP max/avg/min | RMS Error max/avg/min |
|---|---|---|---|
| P1 | 37/34/28 | 1.17/0.85/0.75 | 2.4/1.3/1.0 |
| P2 | 35/27/17 | 2.21/1.24/0.84 | 7.8/4.1/1.4 |
| P3 | 36/27/10 | 4.21/1.32/0.80 | 14.3/5.3/1.24 |
| P4 | 38/34/27 | 1.29/0.97/0.79 | 1.9/1.3/1.0 |

The performance evaluation results of a multi-constellation multi-frequency GNSS receiver in Figure 3 show that the urban canyon imposes greater challenges to positioning and navigation systems, and special measures need to be adopted to improve the GNSS availability and accuracy. The quality of the received signals and the statistical characteristics may vary in changing environmental contexts, and these variations can easily be observed in the received signal power, range measurements (i.e., the path delay and phase difference) and frequency because these factors are the major contributors to the correlation curve between the received signal and receiver-generated replica.

In this paper, the quality of the received signals is analyzed by the Carrier-to-Noise Ratio (CNR) and range residuals along with the elevation angle of satellites. The signal quality assessment tests performed for the aforementioned observation windows, i.e., P1, P2, P3 and P4, are shown in Figure 4. Figure 4a shows the elevation angles of satellites over the entire observation period of 80 min, Figure 4b represents the CNR, and Figure 4c shows the range residuals.

Typically, CNR is influenced by the elevation angle, and therefore PRN: 13 was selected here for quality assessment with increasing elevation angle trends during the observation periods. The signal characteristics in each observation window (i.e., P1, P2, P3 and P4) are highly correlated with the variations in fundamental quality indicators shown in Figure 3.

Both the GNSS fundamental quality indicators and quality of signal are equally influenced by the type of environment. In the P1 and P4 observation windows, the excellent signal strength (i.e., CNR) with great constancy and minimal range residuals corresponds with better GNSS performance, i.e., adequate satellite availability, better DOP values and smaller RMS error due to open-sky views.

Similarly, the signal quality in the P3 and P4 windows exhibited similar characteristics, i.e., significant variations in the signal strength, and higher values of range residuals were highly correlated with variations in the GNSS fundamental quality indicators as shown in Figure 3.

During the observation windows P1 and P4 in Figure 4b, the CNR values showed great constancy due to open-sky views with mostly direct LOS reception. The average CNR values during the P1 and P4 observation windows were found to be 41 and 45.3 dB, respectively. The range residuals for P1 and P4 were almost zero confirming the fact that the environmental context in which the receiver was operating in P1 and P4 was suitable for navigation. In the second observation window, i.e., P2, the signal quality degraded somewhat compared to P1 and P4.

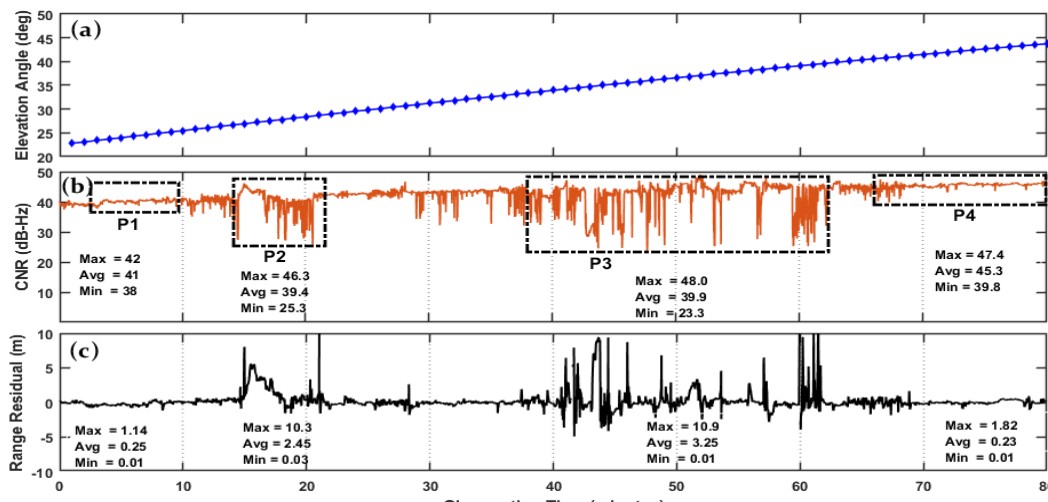

**Figure 4.** The satellite signal quality assessment of GNSS set to quad-constellation mode in different types of environments denoted by P1, P2, P3 and P4 in Sukkur city. (**a**) Elevation angle, (**b**) Carrier-to-Noise Ratio (CNR) and (**c**) Range Residuals.

The CNR values deviated between 25.3 and 46.3 dB. These variations show a significant correlation with the range residuals, as the average and maximum range residuals were found to be 2.45 and 10.3 m, respectively, in Figure 4c. On comparing the range residuals for the observation window P2 with those of P1 and P4, we concluded that these values of range residuals can lead to significant errors in the positioning solution.

Finally, the third observation window (i.e., P3) had the greatest effect on the quality of the received signals, which can be easily identified in Figure 4b,c in terms of the CNR and range residuals, respectively. Although the elevation angle for the P3 window is mostly above 30°, there are rapid random fluctuations in the CNR with a minimum observed value of 23 dB and a maximum value of 48 dB leading to an average range residual of 3.25 m, which is significant when lane-level accuracy is required.

The field experimentation and performance evaluation of multi-constellation and multi-frequency GNSS indicates that, despite remarkable advancements in GNSS-based navigation technologies, standard GNSS services still cannot achieve the required navigation performance across all kinds of environmental contexts. In fact, it is evident from the performance evaluation results that the GNSS receiver experienced severe inaccuracies when navigating from open-sky views to dense urban areas.

Hence, the deployment of additional mechanisms to minimize the inaccuracies is inevitable in order to meet the required navigation performance in such environments. Although there has been substantial research into improving GNSS performance in urban environments by detecting, modeling, de-weighting and mitigating the NLOS/multipath effects at various levels, such as the antenna, receiver and measurement or position levels. These techniques work fairly well in certain environments but are of limited effectiveness in others as discussed in detail in Section 1.

Hence, after the performance evaluation of a GNSS receiver in the different environments above, we concluded that, in order to operate reliably and effectively in a wide range of environments, a GNSS receiver is required to adopt different mitigation techniques according to the detected environmental context. This requires the design and implementation of a model that is adaptive in nature, can accurately detect and identify working environments and can then adjust the mitigation strategy accordingly.

Keeping this in mind, in the next section, a new context-aware navigation (CAN) model is presented that works on an adaptive mitigation strategy based on the detected environment. The proposed method is validated through field experimentation in the subsequent sections.

## 4. Context-Aware Navigation (CAN)

In positioning and navigation, context is the environment that a navigation system is operating in, and this can result in significant effects on the satellite availability and quality of the received signals. Recently, context-aware navigation has received significant attention because it does not require any major hardware modifications and can be easily incorporated into a receiver. As demonstrated in the performance evaluation results, the fundamental quality indicators show a great deal of variations when moving across different environments in urban and suburban areas.

Based on these results, a set of temporal features can be extracted from the GNSS measurements and can be used to distinguish low-to-highly degraded multipath environments. Previously, several algorithms have been proposed for environment detection and characterization using the satellite availability, DOP, residuals and signal strength or its variants [36–42]. In [36], Groves laid the foundation of environment context detection and characterization based on GNSS signal characteristics and measurements.

The preliminary context detection experiments in [36] showed that the signal strength can significantly vary in different environments and that this can be used to distinguish indoor from outdoor environments. Then, in [37], a method was proposed that initially classifies the environment into indoor and outdoor categories and then classifies the outdoor category into open sky and urban environments based on two features extracted from the GNSS measurements, i.e., the satellite availability and signal strength.

Similarly, the satellite availability, height and signal strength were also investigated as context detection features in [38,39]. In [40], the operating environment was categorized as indoor, intermediate and outdoor using the number of detected satellites and variations in signal strength as characterization parameters.

In [41], a five-dimensional signal feature vector (i.e., the mean and standard deviation of the signal strength attenuation, blockage coefficient, geometric dilution of precision (GDOP) and strength fluctuations) was proposed to characterize the operating environment into six categories, such as urban canyon, semi-urban, suburban, viaduct-up, viaduct-down and boulevard. Similarly, ref. [42] proposed a model based on behavior association for further improving the environment detection.

Most of the previous work focused on the development of context-detection models using single and dual constellations with little-to-no consideration regarding how these models can be utilized at the at receiver level to improve the availability and accuracy of navigation services. Furthermore, several researchers have used signal characteristics (i.e., the number of visible satellites, DOP and signal strength/CNR) to propose algorithms for environment detection using machine-learning techniques, fuzzy-inference systems, stochastic modeling and the Hidden Markov model [36–42].

However, the completion of emerging navigation systems (i.e., BeiDou and Galileo) along with the modernization of existing constellations (i.e., GPS and GLONASS) promises significant improvements in satellite visibility, geometry, the quality of navigation signals and, thus, the overall localization accuracy. Furthermore, remarkable advancements in receiver technology are paving the way for using multi-constellation systems.

In almost all of the previous work on context detection, signal strength or its variants were used as the fundamental environment recognition parameters; however, this may not result in accurate context detection in the case of multi-constellation GNSS because: (1) the strength of the received signal is highly affected by NLOS and/or multipath; however, the severity of the effects varies with the frequencies [43]; (2) signal strength is majorly influenced by elevation angle and can also be affected by the receiver efficiency and antenna design; (3) the combination of multiple navigation systems in multi-constellation GNSS mode results in increased satellite density, and therefore monitoring the strength of each satellite can lead to a huge processing load.

Hence, the signal strength may not be considered as an accurate recognition parameter when using multi-constellation and multi-frequency GNSS. In addition, most of the previous models relied on fixed-environment context definitions, giving little or no consideration

to the dynamics of environment. However, considering the on-ground vehicle navigation, the GNSS receiver is not static and continuously encounters changing environmental contexts while navigating through cities.

In dynamic mode, the GNSS receiver experiences a wide range of changing environmental contexts in urban or sub-urban areas, thereby, making context detection extremely difficult. In order to cope up with detecting the rapidly changing environmental context and to improve the positioning accuracy, a new context-aware navigation (CAN) algorithm was proposed that utilizes the globally and equally distributed property of GNSS, i.e., the satellite availability (the average number of satellites in a given geo-location) and a new feature—named the change factor (CF)—which estimates the normalized change in satellite availability in a pre-determined interval of time for decision making. The detailed description of working, implementation and environment characterization of the proposed CAN algorithm along with the working model of the CAN receiver is given in the subsequent sections.

### 4.1. Working of a CAN-Enabled GNSS Receiver

A GNSS receiver consists of a radio frequency (RF) front-end, signal-processing module and a data processing unit. The RF front-end performs amplification, frequency-down conversion and the digitization of a received signal. The digitized signal is then passed to a signal-processing module where acquisition and tracking is performed to recover the navigation message. Finally, in the data processing unit, the navigation message is decoded, and the positioning parameters are estimated.

This is the case of a standard receiver, which is freely available. However, the CAN receiver has the added feature of context detection, which is achieved by forming a recursive closed loop of the position and navigation parameters block with the tracking loop. The position and navigation block passes the positioning parameters to the CAN algorithm block, which characterizes the environment and then adopts a mitigation strategy, which is based upon changing the tracking-loop parameters in order to handle the wider dynamics to counter the multipath effects.

A complete design and working model of the CAN GNSS receiver is shown in Figure 5. The flow-chart of this receiver is given in Figure 6, which explains the working of the CAN receiver in detail.

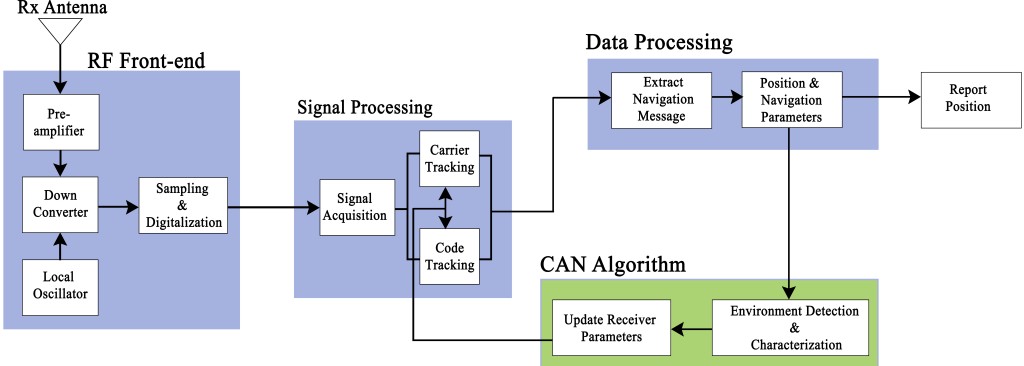

**Figure 5.** A GNSS receiver design based on the context-aware navigation (CAN) algorithm.

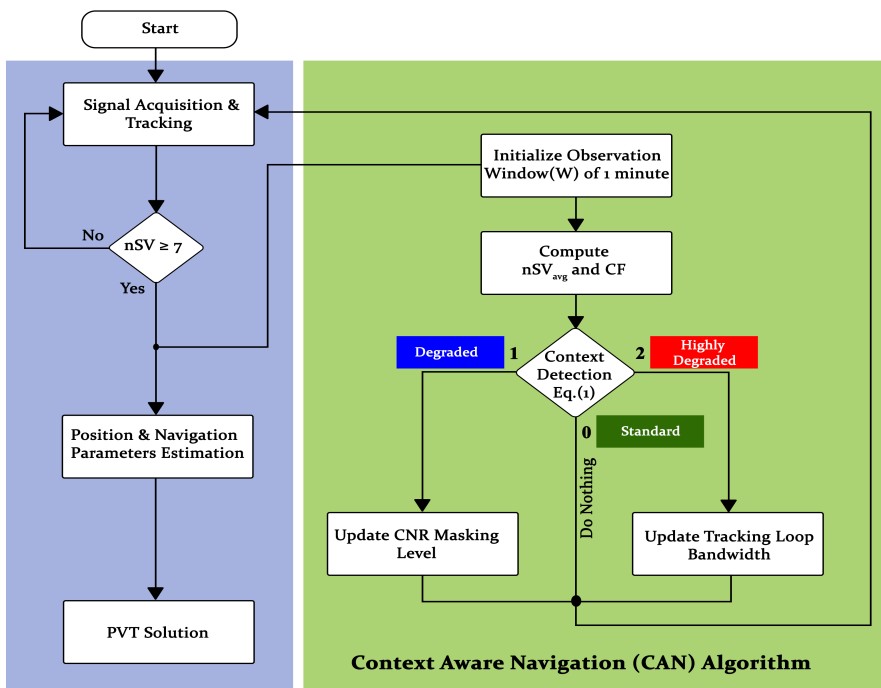

**Figure 6.** A complete workflow of the context-aware navigation (CAN) algorithm, which can detect and characterize the working environment of a GNSS receiver into three categories and then apply an appropriate mitigation strategy as per the detected environment.

The CAN receiver starts by acquiring the signals and then performing tracking of the desired constellations. In the case of the four constellations used in this paper, the minimum number of satellites required for a positioning solution is 7. If the locked satellites are greater than 7, the receiver passes the measurements to the next level for positioning estimation, and the same information is also passed to CAN for initializing the observation windows of 1 min.

In each observation window, the average number of satellites ($nSV_{avg}$) and change factor ($CF$) are computed. The CAN uses $nSV_{avg}$ and $CF$ as the input parameters and then assigns categories to the measurements based on the context definitions given in Equation (1), which classifies the environment into three categories, named standard, degraded and highly degraded, which are encoded as 0, 1 and 2, respectively. The CAN recognizes the environment as standard when $nSV_{avg} \geq 30$ and $CF \leq 0.2$.

The context definition for a degraded environment is $30 < nSV_{avg} \leq 15$ and $0.2 < CF \leq 0.6$ and a highly degraded environment is declared when $nSV_{avg} < 15$ and $CF > 0.6$. Once the working environment of the receiver has been detected and characterized, the CAN algorithm decides whether to update the CNR masking level in the case of a degraded environment or to update the tracking loop noise bandwidth in the case of a highly degraded environment.

In the case of a standard environment, the algorithm does nothing, and the receiver works in its normal mode. It should be noted that we used four constellations to identify the environment; however, the context definition given in Equation (1) may be changed as per the user requirements based on how many navigation systems are opted for using CAN.

$$f(nSV_{avg}, CF) = \begin{cases} 0 \text{ (Standard)} & nSV_{avg} \geq 30, \\ & CF \leq 0.2 \\ 1 \text{ (Degraded)} & 30 < nSV_{avg} \leq 15, \\ & 0.2 < CF \leq 0.6 \\ 2 \text{ (Highly degraded)} & nSV_{avg} < 15, \\ & CF > 0.6 \end{cases} \quad (1)$$

where $nSV_{avg}$ is the average number of satellites in an observation window of 1 min and $CF$ is the change factor computed as

$$CF = \frac{nSV_{max} - nSV_{min}}{nSV_{max}} \qquad (2)$$

where $nSV_{max}$ and $nSV_{min}$ are the maximum and minimum number of satellites tracked in an observation window of 1 min. In most of the previously proposed models, the signal characteristics (i.e., the number of visible satellites, DOP and signal strength/CNR) along with machine-learning techniques, fuzzy inference systems, stochastic modeling and Hidden Markov models [36–42] were used to propose environment-detection models, which is not the case in the proposed CAN algorithm.

The main advantage of using CAN is its simplicity in terms of implementation design, which can be integrated into a GNSS receiver without many hardware modifications. Additionally, the efficient design of the proposed CAN, which includes the selection of optimal GNSS parameters along with context definitions for environment characterization makes it more practically realizable compared to complex methods, which would become practically impossible to implement in the receiver.

*4.2. Environment Detection and Characterization Using CAN*

This section explains the strategy used by CAN to distinguish between different types of environments. In order to validate the performance of CAN in real-world scenarios, a field experiment was performed by enabling CAN in a GNSS receiver in quad-constellation mode. The experiment was performed on the same route and at the same time as used for the performance evaluation study in Section 2.1 above.

Although it is difficult to maintain the exact same trajectory, we attempted to make it as close as possible by driving along the specific lane of the road over the same observation time. It should be noted that the total field experiment time was kept the same (i.e., 80 min) with and without using CAN; however, the times of each observation window (i.e., P1, P2, P3 and P4) and vehicle lane slightly differ from those used previously due to the overall route length, traffic density and vehicle speed.

The context detection along with the categorization results are shown in Figure 7. Figure 7a shows the instantaneous and average number of satellites, Figure 7b represents the change factor (CF) in each observation window, and Figure 7c shows the context classification/categorization. The environment is categorized as standard, degraded and highly degraded based on the parameter values given in Equation (1). If a mitigation strategy is activated using CAN, it will only be initiated for the degraded and highly degraded cases because the standard environment has good accuracy and is taken as a reference to differentiate between the degraded and highly degraded environments.

In Figure 7c, the thick green horizontal line is the region of the standard environment, and the green markers indicate the points when the vehicle environment was characterized as standard by CAN. Similarly, the thick purple horizontal line shows the degraded environment region, and the blue dotted markers indicate the time of the degraded environment, whereas the thick pinkish horizontal line is the highly degraded region, and the red markers indicate the points when CAN declared it as a highly degraded environment. In Figure 7c, from 0 to 16 min, 27 to 34 min and 61 to 77 min, the environment is mostly characterized as standard because the average number of satellites, $nSV_{avg}$, is greater than 30 (Figure 7a), and the CF is less than 0.2 (Figure 7b) during these time slots.

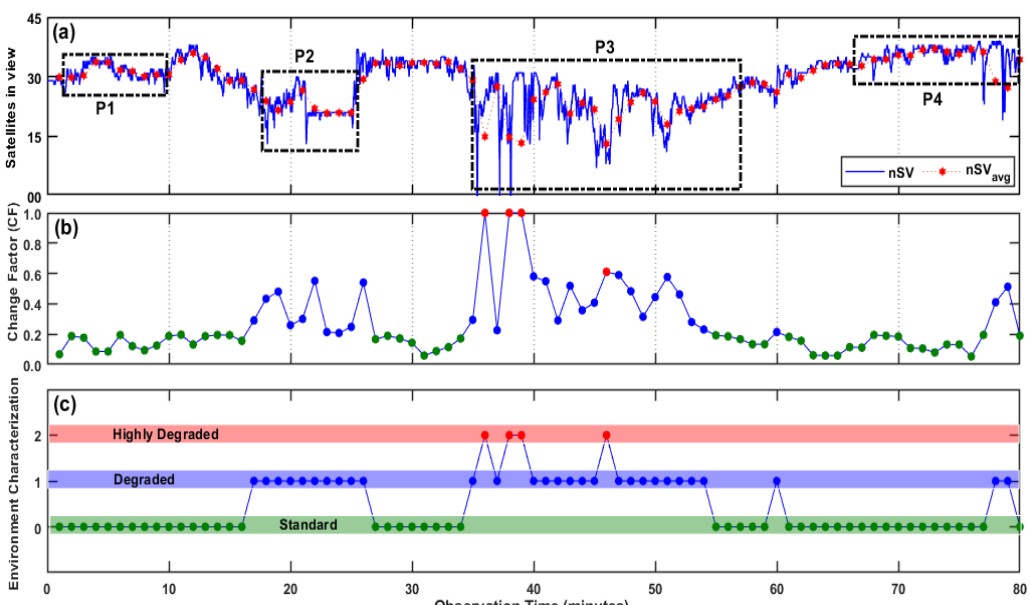

**Figure 7.** Environment context detection and categorization using CAN based on *nSV*$_{avg}$ and *CF* parameter values. (**a**) Satellites in view, (**b**) Change Factor (CF) and (**c**) Environment Characterization.

This means that the moving vehicle may have direct LOS contact with a majority of satellites locked by the receiver, which can lead to good positioning accuracy. On the other hand, between 17 and 26 min, 40 and 45 min and 47 and 60 min and at the 35th, 37th, 78th and 79th min, the environment is characterized as degraded by CAN because, as shown in Figure 7a,b, the average number of satellites, i.e., *nSV*$_{avg}$, falls between 15 and 30 leading to a CF value of greater than 0.2 and less than 0.6, respectively, for most of the time. The degraded environment is usually a semi-urban or urban area where the blockage has a limited effect when using all four constellations.

This is because there always exists a sufficient number of satellites that a GNSS receiver can hold lock on. However, multipath/NLOS still remains a major hurdle in degraded environments for accurate localization and navigation. The CAN algorithm leverages the redundancy of range measurements when using multi-constellation GNSS because it provides the user with a much greater choice of signals and, thus the accuracy can be improved by selecting only those signals that are least contaminated by multipath environments and NLOS to improve the positioning accuracy.

The last case is the environment that is characterized as highly degraded by CAN at some points in P3. This type of environment may be a dense and congested urban area with blockages or high multipath/NLOS signal reception. This is usually the case where the minimum number satellites required for positioning are either inadequate or the available satellites have been affected by multipath/NLOS such that there is no positioning solution or, if the positioning exists, then there may be large errors. When using a multi-constellation GNSS receiver, there is a low probability of facing a highly degraded environment due to a large number of satellites available from different constellations. This is why it can be seen in Figure 7c that the highly degraded environment is identified by CAN at the 36th, 38th, 39th and 46th min in the P3 observation window.

### 4.3. Mitigating Multipath/NLOS Effects in Different Environmental Contexts

The detailed analysis of an on-ground GNSS receiver, characteristics of the received signals and real-time context detection indicate that the severity and nature of degradation is inherently dependent on the type of environment, which is responsible not only for signal degradation but also for MP/NLOs signal reception. Hence, a context detection mitigation strategy can be an effective approach for mitigating the multipath effects. In this

regard, CAN can be a useful tool that can be employed inside a GNSS receiver to counter multipath effects by adopting an adaptive mitigation strategy based on context detection.

In this section, the utilization of CAN to mitigate multipath effects for improving the positioning accuracy is demonstrated. The environment categories in CAN, i.e., dtandard, degraded and highly degraded, are the indication of levels of deterioration effects, i.e., blockage, MP and/or NLOS. The mitigation strategy adopted here will be initiated for degraded and highly degraded cases only because there are minimal chances of performance degradation in a standard environment.

Typically, in a degraded environment, the impact of blockage is found to be minimal because there is always a sufficient number of satellites (i.e., >15) that a GNSS receiver can acquire and track, and this can be verified by analyzing the P2 and P3 windows in Figures 3a and 7c, respectively. However, in such environments, the multipath/NLOS reception remains the major contributor to inaccuracies because the overall positioning solution depends on the number of tracked satellites and the quality of the received signals.

Therefore, the proposed CAN method leverages the redundancy of range measurements when using multi-constellations because it provides the user with a much greater choice of signals, and thus the accuracy can be improved by selecting only those signals that are least contaminated by multipath environments and NLOS to improve the positioning accuracy. There exist several studies that have demonstrated the improvements in the positioning accuracy by detecting, de-weighting or excluding faulty range measurements [29,55–60].

Conventionally, sequential monitoring is used to rank and discard the affected measurements based on range residuals, the elevation angle and the CNR level. However, in the case of multi-constellation mode, individually monitoring a huge number of satellites with several parameters will result in a greater processing load and may cause a delay in decision making. Keeping this limitation in-view, CAN uses an adaptive approach based on the detected environment instead of individually monitoring and discarding the satellites. The signal strength or CNR level is a clear indication of the signal reception conditions, i.e., LOS or MP/NLOS reception.

Typically, the satellites with high CNR values result in more accurate positioning solutions because these are less likely to be affected by MP/NLOS. In GNSS receivers, the CNR masks can be used to improve the accuracy by selecting signals with high CNR and excluding signals that have been attenuated due to reflection or diffraction. In the default settings, the CNR mask level is kept as low as 10 dB as shown in Table 5; however, when a degraded environment is detected by CAN, the CNR level is increased to 35 dB in order to discard the MP-affected signals.

On the other-hand, the mitigation strategy adopted by CAN completely changes in the case of a highly degraded environment. Instead of discarding the satellites affected by MP, CAN updates the delay locked loop (DLL) bandwidth to deal with wider dynamics. The receiver settings used by CAN for mitigating multipath effects for different types of environments are shown in Table 5. The standard mode parameters are the default settings used for tracking the incoming signals. For a degraded environment, only the CNR masking level is increased whereas, for the highly degraded case, the CNR mask level is kept at the default value while the DLL bandwidth is increased.

**Table 5.** The GNSS receiver level settings in three different operating contexts.

| Receiver Settings | Environmental Contexts | | |
|---|---|---|---|
| | Standard | Degraded | Highly Degraded |
| DLL Bandwidth (Hz) | 0.25 | 0.25 | 01 |
| CNR Mask (dB-Hz) | 10 | 35 | 10 |
| Elevation Mask | 10° | 10° | 10° |
| PLL Bandwidth (Hz) | 15 | 15 | 15 |
| Data Rate (Hz) | 01 | 01 | 01 |

In previous works, the fixed CNR masks or elevation masks were used for the detection and exclusion of faulty measurements, such as signals affected by MP or NLOS, to improve the positioning accuracy. These mitigation models or techniques, however, work effectively only in some contexts and not in all operating environments. The adaptive design of the proposed CAN allows the receiver to adaptively mitigate the effects of multipath or NLOS by updating the receiver tracking-loop parameters based on the inferred environment.

The performance of a GNSS receiver in quad-constellation mode (i.e., GPS, GLONASS, Galileo and BeiDou) in terms of fundamental quality indicators with CAN enabled are shown in Figure 8. Figure 8a shows the number of tracked satellites, Figure 8b represents the position dilution of precision (PDOP), while the RMS error over the entire observation period is shown in Figure 8c. The results in Figure 8 show that almost the same level of performance was achieved in the observation windows P1 and P4 (standard environment cases) when compared with the results of the field experiments performed in Figure 3 without using CAN.

This is clear because the CAN algorithm does not initiate any mitigation strategy when working in standard environments as explained above. On the other hand, in P2 observation window, the average number of satellites dropped to 22 in Figure 8a, which were 27 in Figure 3a without using CAN. In the P2 observation window, the environment was mostly identified by CAN as degraded, and thus the dropping of satellites is because of setting a higher CNR mask level and, thus, discarding the possible multipath/NLOS measurements in order to avoid them being used in the positioning solution.

The effects of excluding the contaminated measurements in P2 using CAN can be seen in Figure 8c, where the average RMS error dropped to 3.61 m, which was 4.14 m without using CAN in Figure 3c. On the other-hand, the PDOP values in Figure 8b increased slightly with an average of 2.17, which was 1.24 without using CAN in the same observation window. Relying on PDOP alone is not a good indicator to check the efficacy of a positioning solution because, when there are multipath measurements, a good PDOP value may be attained along with significant errors in the positioning solution.

This is what we experienced after applying CAN and, thus, excluding the multipath/NLOS measurements in the P2 observation window leading to increased PDOP values but with improved positioning accuracy. In the P2 observation window, the sole purpose of excluding the faulty measurements is to improve the positioning accuracy, which was successfully achieved after initiating CAN. In observation window P3, the environment is a blend of both the degraded and highly degraded environments due to rapid variations in the environmental context as a result of the moving vehicle.

It is the most difficult to apply mitigation models to this type of route as the environmental context changes in a matter of seconds, e.g., the receiver may be working in one environment to apply the mitigation strategy when the context changes to another. In order to effectively counter this problem, the CAN algorithm uses 1 min intervals for estimating the parameter values (i.e., $nS_{avg}$ and CF) for environmental characterization and then initiates the mitigation strategy accordingly. In the P3 observation window, the maximum RMS error was found to be 7.31 m as shown in Figure 8c, which was 14.3 m without using CAN in the same observation window as can be seen in Figure 3c.

Similarly, the average RMS error using CAN dropped to around 2.93 m compared to 5.3 m without using CAN. This is a large improvement in the positioning accuracy. The PDOP values are almost same in the observation window P3 with and without using CAN. our comparison of the parameters used for the performance evaluation of CAN in different environments is summarized in Table 6 below. The table compares the parameters, including the satellites in-view, PDOP values and RMS errors.

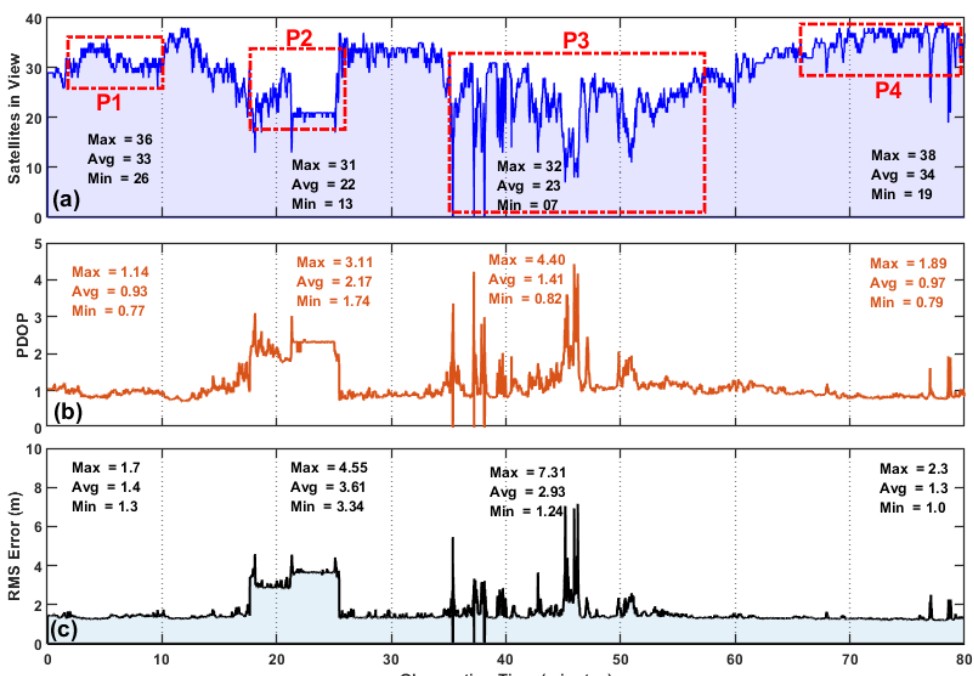

**Figure 8.** The performance evaluation results of a CAN-enabled GNSS receiver in quad-constellation mode in terms of fundamental quality indicators, i.e., (**a**) satellite availability, (**b**) PDOP and (**c**) RMS error over the entire experimental route containing different operating environments.

**Table 6.** Comparison of the quad-constellation GNSS receiver performance with CAN enabled under different types of environments denoted by P1, P2, P3 and P4 observation windows.

| Site | Satellites in View max/avg/min | PDOP max/avg/min | RMS Error max/avg/min |
|---|---|---|---|
| P1 | 36/33/26 | 1.14/0.93/0.77 | 1.7/1.4/1.3 |
| P2 | 31/22/13 | 3.11/2.17/1.74 | 4.44/3.16/3.34 |
| P3 | 32/23/07 | 4.40/1.41/0.82 | 7.31/2.93/1.24 |
| P4 | 38/34/19 | 7.31/2.93/1.24 | 2.3/1.3/1.0 |

The degraded and highly degraded environments are the ones most affected by the multipath/NLOS signal reception resulting in reduced positioning accuracy. In the case of a highly degraded environment, the range errors were either large or there were not sufficient number of satellites available to estimate the position resulting in no position at all. Since the P3 observation window contains both the degraded and highly degraded environmental contexts, the P3 observation window was taken here as a case study to compare the overall performance of a dynamic mode GNSS receiver in achieving lane-level accuracy using a quad-constellation configuration with and without using CAN by considering the total number of positioning points during the P3 observation window.

As mentioned earlier, the P3 observation window is a densely populated area with high-rise buildings, congested roads, flyovers and two cantilever bridges (Lansdowne Bridge). The window spans over 25 min (37–62 min) in Figure 3 without using CAN and is 22 min (36 to 58 min) when using CAN as can be seen in Figure 8. There are approximately 1500 measurement points in the P3 observation window if we take 25 min as the complete observation time in P3 with and without using CAN.

The number of occurrences of RMS errors in the P3 observation window on a scale of 1 to 10 m is shown in Figure 9. In the case of CAN, more than half of the RMS errors fall within 2 m—consisting of almost 800 points out of 1500 measurement points. On the other hand, without initiating CAN, only 489 points out of 1500 measurement points suffered

errors less than 2 m while more than 1000 measurements suffered errors greater than 3 m, which is approximately 16 min out of the total route length of 25 min.

This is further elaborated in a clearer way by using a pie-chart in Figure 10, which shows the number of occurrences of RMS errors greater than 2 m and less than 2 m. The pie-chart shows the overall performance of the multi-constellation and multi-frequency GNSS receiver in dynamic mode with and without using CAN. The RMS error was found to be less than 2 m for 53% of the total experimental period with CAN and 32% without using the proposed CAN in the P3 observation window.

This improvement in the positioning accuracy demonstrates the efficacy of CAN in degraded and highly degraded environments. We concluded that the CAN method is effective for mitigating the multipath effects in order to improve the positioning accuracy and availability in the case of degraded and highly degraded environments without using any aiding devices or additional hardware.

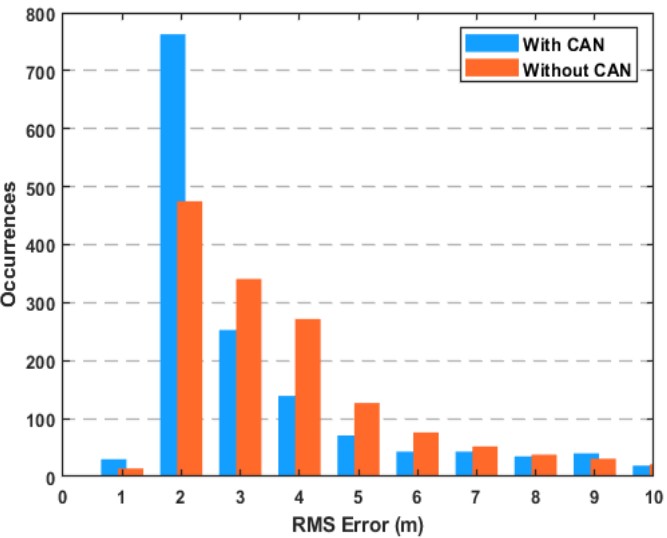

**Figure 9.** GNSS performance in terms of the RMS error at P3 with and without CAN.

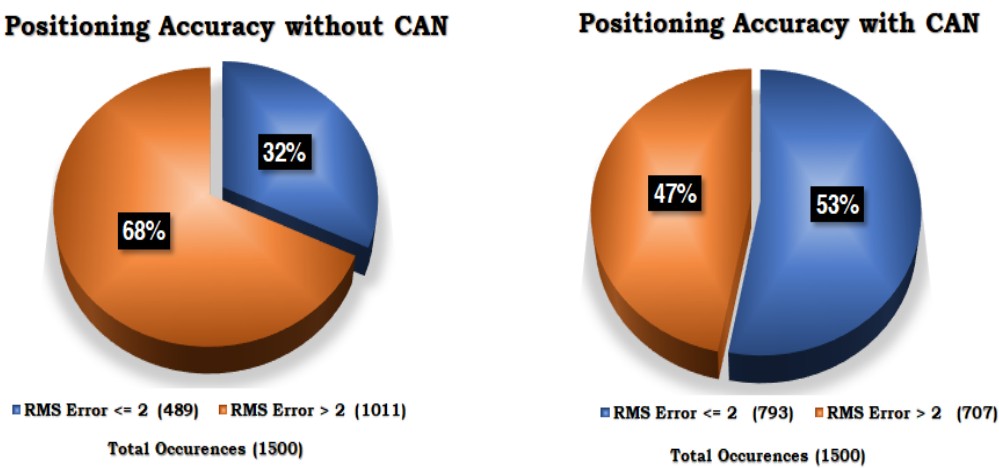

**Figure 10.** GNSS accuracy achieved in a highly degraded environment (P3) with and without CAN.

## 5. Conclusions and Recommendations

The GNSS signal characteristics provide advantages in identifying and characterizing the environmental context to improve the positioning accuracy. This paper reinforced the fact that multi-constellation GNSS performance varies greatly with the environmental context and is highly likely to deteriorate in dense urban environments surrounded by high-rise buildings, congested roads, flyovers and under-passes. A performance evaluation

field study in these complex environmental scenarios revealed that a multi-constellation GNSS receiver, even after using four constellations, lacked consistency in maintaining the required accuracy thresholds across all kinds of environments.

The significant deviations in the number of tracked satellites, geometric distribution of satellites (PDOP) and quality of the navigation signal in the degraded and highly degraded environments due to signal obstructions and NLOS/multipath effects resulted in erroneous, inconsistent and unreliable positioning solutions. To increase the capability of a multi-constellation GNSS receiver, the context-aware navigation (CAN) method effectively utilized the multi-constellation GNSS receiver measurement parameters to first identify the type of working environment and then initiated a mitigation strategy accordingly for optimal receiver performance.

This was achieved by CAN through classifying the environment into three distinct categories: standard, degraded and highly degraded, based on two estimated features. After environment detection and characterization, CAN initiated the mitigation strategy by updating the CNR masking level in the case of a degraded environment or by changing the receiver tracking-loop parameters in the case of a highly degraded environment. The advantage of using CAN is its simplicity in terms of the implementation design, as it can be integrated in a GNSS receiver without great modifications.

The efficacy of CAN was validated in this paper through field experimentation on the same route and similar environment dynamics as used for a GNSS receiver without using CAN in a quad-constellation mode. The real-time implementation of CAN navigating through the different types of environmental contexts showed that this algorithm reduced the signal degradation effects in degraded and highly degraded environments by improving the lane level accuracy, which was achieved 53% of the time when using a CAN-based GNSS receiver and was achieved only 32% of the time when not using CAN.

In upcoming work, this study will be extended to larger context ranges, higher analysis windows, complex physical multipathways and concentrated quad-constellation conditions. Moreover, a more robust CAN algorithm using Deep-Markov inference will be designed to not only detect and classify an appropriate context but also to make autonomous variational decisions over the available set of GNSS signals. Furthermore, comparative analysis will be made between the offline and online compensation techniques for benchmark parameters, settings and configurations of the receiver under distinct dynamic environments.

**Author Contributions:** Conceptualization, A.H., A.A. and M.A.S.; methodology, A.H. and M.A.S.; software, A.H. and A.A.; validation, A.H., M.A.S. and S.K.; formal analysis, S.K.; investigation, A.H.; resources, M.A.S. and L.S.; data curation, A.H.; writing—original draft preparation, A.H.; writing—review and editing, A.H., L.S. and H.M.; visualization, S.K., L.S. and H.M.; supervision, A.A.; project administration, M.A.S.; funding acquisition, A.A. All authors have read and agreed to the published version of the manuscript.

**Funding:** The authors wish to acknowledge the HEC (Higher Education Comission) of Pakistan for providing the financial support under National Research Program for Universities (NRPU) grant 6250/Sindh/NRPU/R&D/HEC/2016 for this research work.

**Institutional Review Board Statement:** Not applicable.

**Informed Consent Statement:** Not applicable.

**Data Availability Statement:** Not applicable.

**Conflicts of Interest:** The authors declare no conflict of interest.

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
