# Peer review of "On Mitigating the Effects of Multipath on GNSS Using Environmental Context Detection"

_applsci, doi:10.3390/app122312389_

Round 1

Reviewer 1 Report

The paper deals with an important topic and it is nicely prepared. The authors presented their work in a concise and clear manner. I have only a few small suggestions given below.

1. Lines 84-85 - There are few typing errors ("nnother", "mulitpath"); please rectify.

2. Line 138 - Acronyms PDOP and RMS are mentioned for the first time in the text but not written in full; please rectify.

3. Line 397 - Typing error: "mintue"; please rectify.

Author Response

Original Manuscript ID: applsci-1888534         

Original Article Title: “On Mitigating the Effects of Multipath on GNSS using

Environmental Context Detection”

To: Applied Science Reviewer

Re: Response to reviewer

Dear Reviewer,

Thank you for allowing a resubmission of our manuscript, with an opportunity to address the reviewers’ comments.

We are uploading (a) our point-by-point response to the comments (below) (response to reviewers), (b) an updated manuscript with yellow highlighting indicating changes, and (c) a clean updated manuscript without highlights (PDF main document). Please see the attachment.

Best regards,

Arslan Ahmed

Reviewer 2 Report

The paper is written in proper English, it is pleasant to read. The introduction is informative, followed by appropriate citations. It provides a good background to both professionals and younger parties interested in the aforementioned topic of GNSS, PNT, and related aspects in various environmental conditions, applications, etc. However, there are several issues that need to be properly addressed.

Suggestions and comments:

Minor editorial and formatting issues are present, e.g., lack of space or multiple (unnecessary) space signs between subsequent words, etc.

Before describing the laboratory setup, Authors should point out and highlight the novelty of their paper. Additionally, do discuss the organization of this paper, including what is given in each Chapter/Section.

What kind of antenna(s), receiver(s), as well as additional equipment for signal processing, was utilized during your study? What was its technical specification? What was the duration of your study? Was it carried out during a single day? Or during one week, one month? What was the height at which the antenna was mounted on the roof of the car? A table, followed by appropriate description, would seem necessary.

Additionally, what frequency bands were you able to check and monitor? This strictly depends on the technical specs of the utilized equipment.

What about other systems, like mobile cellular telephony, etc., operating in the GHz satellite frequency bands? Were there any interruptions caused by such systems? Did they or could they influence your measurements? Additional comments seem necessary.

Moreover, why did you select these particular routes? What was the characteristics of their surroundings? Did it include dense urban buildings, multi-story buildings, etc., or was it close to open terrain, with trees, parks, etc. How many loops (iterations) did you take along each route? This information is significant, yet it is missing.

When planning a research campaign, one can predict the number of observed (and monitored) satellites. Just by using software or webpages, that enable to obtain this information. So, what was the number of expected observed (monitored) satellites, and how many of the did you really discover?

Next, when planning such a campaign, it is necessary to give a graphical representation of the planned route, than compare it with the measured route (based on obtained GNSS measurements). Additional maps describing the urban fabric, real route, measured (obtained with errors) route, etc., are necessary.

When talking about errors, in what planes do you describe them? In the Vertical or Horizontal one? What was the actual vs measured level (height) above sea level? How did the equipment perform in case of, e.g., straight lines, and taking turns, etc.?

Currently, the Conclusions section is too short and not convincing at all. What about Future studies and open aspects. Provide additional feedback for the potential reader. Do discuss possible directions for both your own as well as other upcoming investigations.

The list of cited References lacks interesting papers focused on, e.g., urban navigation using mobile devices, QoS/QoE aspects in satellite/GNSS communications, including surveys, field-test studies, and other investigations. Therefore, Authors are encouraged to look for additional papers and conference proceedings focused on this topic.

This paper has a potential to be a good one. However, currently it has a big number of issues and lacks that need to be properly addressed. Authors should prepare an extended version of their manuscript and correct all imperfections.

Author Response

(The authors gave the same response as above.)

Reviewer 3 Report

1. The innovation

In this paper, CAN algorithm is used to reduce the multipath influence, but there are similar methods in this field to reduce the multipath influence. Please explain the difference between your proposed CAN algorithm and other papers and point out the innovation and advantages of your proposed algorithm compared with other similar algorithms.

The introduction of this paper does not clearly demonstrate the innovativeness of the method compared to traditional algorithms. A comparison of the proposed algorithm with traditional algorithms should be added to clarify the advantages of the proposed algorithm.

And the analysis and derivation of the algorithm in the whole text is very simple, there are only two formulas, the focus is on the application of the CAN algorithm on the receiver and the test of its effect. The advantages of this algorithm are not clear.

2. Methods described

In the introduction of this paper, the can algorithm is not fully described, and its formula is not well reflected, and the description is too simple.

The description of the relevant characteristics of Figure 3 and Figure 4 mentioned in lines 256-257 is too simplistic.

3. Rules of English Writing 

There are some problems in English writing. For example, the "," after "because" in line 120 should be deleted, and the "i.. e" in line 269 should be changed to "i.e.".

4.the picture and text description is wrong

230 lines of data description about Figure 3 is wrong and the data description part of Table 2 about Figure 3 is wrong.

Table4 table format is wrong.

5. Inconsistent Context Semantics

The conclusions expressed in lines 19-21 differ from those expressed in lines 569-571.

Author Response

(The authors gave the same response as above.)

Round 2

Reviewer 2 Report

Thank you for addressing to my suggestions and comments. Provided answers are comprehensive and properly justified. The paper is interesting, it has a practical character. The revised version of this manuscript has improved its overall quality. Therefore, I do recommend it to be accepted and published in the Journal.

Author Response

We are really grateful to the potential reviewer for spending his lot time to point out the shortcomings in the paper that ultimately helped us in improving the quality of the paper.

Once again Thank you very much.

Author Response

We are really grateful to the potential reviewer for spending his lot of time to point out the shortcomings in the paper that ultimately helped us in improving the quality of the paper.

Once again Thank you very much.

Round 3

Reviewer 3 Report

Accepted in its current form.